# Trifuhalol A Suppresses Allergic Inflammation through Dual Inhibition of TAK1 and MK2 Mediated by IgE and IL-33

**DOI:** 10.3390/ijms231710163

**Published:** 2022-09-05

**Authors:** Sim-Kyu Bong, No-June Park, Sang Heon Lee, Jin Woo Lee, Aaron Taehwan Kim, Xiaoyong Liu, Sang Moo Kim, Min Hye Yang, Yong Kee Kim, Su-Nam Kim

**Affiliations:** 1Natural Products Research Institute, Korea Institute of Sceience and Technology (KIST), Gangneung 25451, Korea; 2Division of Bio-Medical Science and Technology, KIST School, University of Science and Technology, Seoul 02792, Korea; 3Department of Food Science, University of Massachusetts, Amherst, MA 01003, USA; 4Haizhibao Deutschland GmbH, Heiliggeistgasse 28, 85354 Freising, Germany; 5Department of Marine Food Science and Technology, Gangneung-Wonju National University, 7 Jukheon-gil, Gangneung 25457, Korea; 6College of Pharmacy, Pusan National University, Busan 46241, Korea; 7College of Pharmacy, Sookmyung Women’s University, Seoul 04310, Korea

**Keywords:** trifuhalol A, allergic inflammation, interleukin-33, immunoglobulin E, degranulation, TGFβ-activated kinase 1, MAPK-activated protein kinase 2

## Abstract

The activation and degranulation of immune cells play a pivotal role in allergic inflammation, a pathological condition that includes anaphylaxis, pruritus, and allergic march-related diseases. In this study, trifuhalol A, a phlorotannin isolated from *Agarum cribrosum*, inhibited the degranulation of immune cells and the biosynthesis of IL-33 and IgE in differentiated B cells and keratinocytes, respectively. Additionally, trifuhalol A suppressed the IL-33 and IgE-mediated activation of RBL-2H3 cells through the regulation of the TAK1 and MK2 pathways. Hence, the effect of trifuhalol A on allergic inflammation was evaluated using a Compound 48/80-induced systemic anaphylaxis mouse model and a house dust mite (HDM)-induced atopic dermatitis (AD) mouse model. Trifuhalol A alleviated anaphylactic death and pruritus, which appeared as an early-phase reaction to allergic inflammation in the Compound 48/80-induced systemic anaphylaxis model. In addition, trifuhalol A improved symptoms such as itching, edema, erythema, and hyperkeratinization in HDM-induced AD mice as a late-phase reaction. Moreover, the expression of IL-33 and thymic stromal lymphopoietin, inflammatory cytokines secreted from activated keratinocytes, was significantly reduced by trifuhalol A administration, resulting in the reduced infiltration of immune cells into the skin and a reduction in the blood levels of IgE and IL-4. In summarizing the above results, these results confirm that trifuhalol A is a potential therapeutic candidate for the regulation of allergic inflammation.

## 1. Introduction

Allergic inflammation is an important pathological condition that includes anaphylaxis, allergic asthma in the lung, atopic dermatitis (AD) in the skin and several allergic diseases involving the nose and eyes [1]. The initial phase of an allergic reaction occurs primarily in seconds or minutes after exposure to allergens. This response is induced by the production of various inflammatory mediators such as cytokines, leukotrienes and prostaglandins by mast cells, as well as the release of histamine and granular proteins by the degranulation of mast cells after the crosslinking of allergen-specific immunoglobulin E (IgE) molecules to FcεRI receptors on mast cells [2]. These mediators cause itching in nerve cells, contractions in the smooth muscle cells of the bronchial airway, mucus production in goblet cells, and vasodilation and edema in endothelial cells [1,3,4].

The cells that mediate the late phase of allergic reactions are mast cells. The mediators of the early-phase recruit leukocytes from the blood to the allergic site and activate them. A high portion of infiltrating lymphocytes observed at the site of an allergic reaction are eosinophils. The cytokines produced by the recruited eosinophils and the T helper 2 (Th2) cells lead to the further infiltration of eosinophils and mast cells and the class switching of B cells to produce IgE. This IgE can bind to the FcεRI receptors of mast cells and prime them, promoting further allergic reactions [5]. The degranulation of mast cells can be mediated by allergic or pseudoallergic routes. IgE triggers classical FcεRI receptor-mediated allergic reactions, and interleukin 33 (IL-33) elicits the MAS-related G protein-coupled receptor-X2-mediated pseudoallergic reaction [6].

Brown seaweed possesses abundant secondary metabolites, which exhibit anticancer, antiviral, antioxidant, anti-inflammatory, cholesterol-reducing, and antidiabetic activities. Such bioactive compounds include phytosterols, polysaccharides, phloroglucinols, phlorotannins, terpenes, fatty acids, and halogenated compounds such as carotenoids [7]. A high level of phlorotannin, a phenolic compound, is present in brown algae, and has been found to have strong anticancer, anti-inflammatory, antioxidant, and UV-protective activities [8]. *Agarum cribrosum* is an abundant and deciduous brown macroalgae distributed in Korea and the Pacific Ocean. However, *A. cribrosum* is not often used for human consumption or in medicine. An acetone extract and its solvent fractions from *A. cribrosum* showed antioxidant, collagenase inhibitory, elastase inhibitory, and tyrosinase inhibitory activities. Fucoidans, the sulfated fucans isolated from *A. cribrosum,* exhibited immunomodulatory activity, and trifuhalol A exhibited anti-inflammatory activity in RAW 264.7 macrophages. However, there is little information on the efficacy of compounds derived from *A. cribrosum* on allergic inflammation [9,10,11].

We found that trifuhalol A inhibits the degranulation of mast cells and IL-33 and IgE production through keratinocytes and differentiated B cells, respectively. Based on these results, trifuhalol A can be efficiently used to treat allergic inflammation. In this study, we evaluated the efficacy of trifuhalol A on allergic inflammation using the Compound 48/80-induced systemic anaphylaxis model for pseudoallergic reactions and the house dust mite (HDM)-induced AD model for allergic reactions using appropriate mouse species.

## 2. Results

### 2.1. Effects of Trifuhalol A on Allergic Inflammation-Related Mediators

To evaluate the potential of trifuhalol A on allergic inflammation, first, a degranulation assay was performed using RBL-2H3 cells. Previously, dinitrophenyl (DNP) was shown to enhance the release of β-hexosaminidase from granules of IgE-sensitized RBL-2H3 cells [12]. DNP-specific IgE treatment significantly increased β-hexosaminidase release, whereas treatment with trifuhalol A inhibited degranulation in a dose-dependent manner (Figure 1a), in which trifuhalol A ameliorated allergic inflammation.

Next, we examined the effect of trifuhalol A on the expression of IL-33, which is related to pseudoallergic or allergic reactions. HaCaT cells were treated with 2 μM of LL-37 and irradiated with ultraviolet B (UVB) at 15 mJ/cm^2^ after 24 h, followed by further incubation for 8 h to measure mRNA expression. The expression of IL-33 was strongly increased in the LL-37/UVB-treated group compared to the nontreated control group at the mRNA level. However, treatment with trifuhalol A reversed the effect of LL-37/UVB on IL-33 expression (Figure 1b). These results suggest that trifuhalol A may improve allergic reactions via IL-33 regulation.

To identify the underlying mechanism of trifuhalol A on IgE-mediated allergic reactions, we measured the production of IgE by U266B1 cells. The IgE levels in the cell culture supernatants stimulated with LPS and IL-4 in the presence or absence of trifuhalol A for 72 h were measured with ELISA. Trifuhalol A significantly decreased IgE secretion in a concentration-dependent manner (Figure 1c), suggesting that trifuhalol A can block the class switch for B cells to produce IgE-type antibodies. Furthermore, the increased expression of IL-4 triggered allergic progression in RBL-2H3 cells, confirming that trifuhalol A can block their priming for further allergic reactions. The treatment with trifuhalol A in the presence of PMA/ionomycin (PI) suppressed the PI-induced increase in IL-4 mRNA expression compared to the control group (Appendix A). However, the effect of trifuhalol A on the expression of IL-4 seemed to be weaker than its inhibitory effect on other mediators, such as IL-33 and IgE. Degranulation in RBL-2H3 cells did not occur with IL-33 treatment, but a combination with IgE showed a slightly greater increase than the application of IgE alone. Trifuhalol A administration reduced the degranulation increased by co-treatment with IL-33 and IgE (Figure 1d). Therefore, these results suggest that trifuhalol A can control both the allergic and pseudoallergic routes in the early and late phases of allergic reactions through the regulation of IgE and IL-33.

### 2.2. Effects of Trifuhalol A on IL-33- and IgE-Mediated Immune Cell Activation

Given that trifuhalol A inhibits the production of IL-33 and IgE and reduces degranulation by them, we next studied the effects of trifuhalol A on NF-κB- and MAPK-activated protein kinase 2 (MK2) in RBL-2H3 cells. IL-33 and IgE augmented the phosphorylation of TGFβ-activated kinase 1 (TAK1) and MK2, leading to the activation of the NF-κB pathway to produce several cytokines, such as IL-13, IL-6 and TNF-α. Trifuhalol A reduced the IL-33- and IgE-induced phosphorylation of TAK1 and MK2 in RBL-2H3 cells (Figure 2a,b). In addition, a reporter gene assay for NF-κB demonstrated that trifuhalol A did not affect the transcriptional activity but suppressed the transactivation of NF-κB evoked by IL-33 and IgE (Figure 2c). NF-κB regulated cytokines associated with TAK1 and MK2—such as IL-13, IL-6 and TNF-α, which were also increased by IL-33 and IgE co-treatment—and their up-regulated transcriptional and translational level was declined by trifuhalol A (Figure 2d–h). These results explain that the suppression of the IL-33- and IgE-mediated activation of RBL-2H3 cells by trifuhalol A can be attributed to the inhibition of the TAK1 and MK2 pathways.

### 2.3. Effect of Trifuhalol A on Systemic Anaphylaxis and Itching

To verify the effects of trifuhalol A on systemic anaphylactic reactions, we used an animal model stimulated with Compound 48/80, a nonimmunologic chemical stimulant. The mice injected intraperitoneally with Compound 48/80 (10 mg/kg) were monitored every 5 min for 1 h to determine the mortality rate. The survival rate of the group with no treatment was 100%, and the survival rate of the Compound 48/80-treated group decreased to 33.3%, whereas the group pretreated with trifuhalol A (200 mg/kg) for 1 h had a 66.7% survival rate (Figure 3a), indicating that trifuhalol A can reduce the mortality rate due to systemic anaphylaxis caused by drug exposure, such as Compound 48/80, by half.

It is known that Compound 48/80 induces itching at a low dose (50 μg/kg) by promoting the secretion of mediators such as histamine and substance P from mast cells [13,14]. To evaluate the effects of trifuhalol A on pruritus, a low dose of Compound 48/80 was used to induce itching in ICR mice. The Compound 48/80-treated group showed markedly increased scratching behavior compared to the control group. However, scratching behavior provoked by Compound 48/80 was significantly ameliorated by trifuhalol A (20 mg/kg) administration (Figure 3b). Taken together, these results show that trifuhalol A ameliorates drug-induced mast cell-mediated systemic anaphylaxis and pruritus.

### 2.4. Effect of Trifuhalol A on HDM-Induced AD-like Lesions

To determine whether trifuhalol A affects the late phase of allergic reactions, we examined its efficacy on atopic dermatitis (AD), a late-phase allergic reaction. The major clinical manifestations emerged after HDM antigen, a well-known allergen, was administered to the back skin of NC/Nga mice. The continuous and repeated application of HDM to the skin for 3 weeks exacerbated the AD-like lesions, inducing skin dryness followed by edema, mild hemorrhage, and erythema, whereas trifuhalol A treatment prevented these unpleasant symptoms (Figure 4a).

To evaluate the dermatitis score, we graded the symptoms of the AD-like skin lesions, such as excoriation/erosion, edema, scarring, and erythema/hemorrhage. The symptoms worsened over three weeks in the HDM group. However, in the trifuhalol A-treated group, the dermatitis score was lower than that in the control group at three weeks than in the HDM group (Figure 4b). The topical application of HDM strongly increased the frequency of scratching by six-fold compared to the control group, and the application of trifuhalol A significantly decreased scratching behavior (Figure 4c).

To further investigate the effects of trifuhalol A on the anti-inflammatory activity in the skin, such as normalizing epidermal hyperplasia and the mass infiltration of immune cells, the skin tissue was removed, and a histological examination was conducted. Based on the results of the H&E staining analysis, trifuhalol A administration ameliorated the harmful epidermal signs, such as an increase in epidermal thickness, due to the hyperproliferation of keratinocytes (Figure 4d,e). Furthermore, a toluidine blue staining analysis proved that the trifuhalol A-treated group had strongly reduced mast cell infiltration into the dermis (Figure 4f,g). These results indicate that trifuhalol A effectively suppressed HDM-induced allergic inflammation-associated signs in NC/Nga mice.

### 2.5. Effects of Trifuhalol A on Serum and Tissue Factors Associated with Allergic Inflammation in HDM-Induced AD-like Mice

The application of HDM causes an increase in serum IgE and IL-4 production in rodent allergy models [15]. According to the results of the serum test, IgE and IL-4 levels increased in the HDM-stimulated group. Hence, trifuhalol A application noticeably attenuated serum IgE (Figure 5a) and IL-4 (Figure 5b) concentrations in HDM-induced AD mice.

Atopic dermatitis induces various reactions in the immune system and skin cells. Activated keratinocytes secrete inflammatory cytokines such as IL-33 and thymic stromal lymphopoietin (TSLP), which increase the infiltration of immune cells such as eosinophils and induce itching. In addition, they secrete a large number of cytokines, such as Th2 cytokines [16]. 

To investigate the effects of trifuhalol A on IL-33 and TSLP protein expression in the HDM-treated dorsal skin of mice, immunohistochemical staining with the appropriate antibodies was performed. The expression levels of IL-33 and TSLP obviously increased in the back skin of mice with atopic dermatitis induced by HDM treatment. However, in the back skin of mice treated with trifuhalol A, the expression patterns of IL-33 and TSLP were almost identical to those of the control group (Figure 5c,d). Next, the gene expression levels of Th2 cytokines, such as IL-4 and IL-13, were measured in skin tissues. The mRNA expression levels of IL-4 and IL-13 were noticeably enhanced in the HDM group measured against the control group, whereas trifuhalol A application significantly decreased Th2 cytokine gene expression compared to the HDM group (Figure 5e,f). Therefore, it is concluded that trifuhalol A ameliorates allergic inflammation by reducing the expression of IL-33, TSLP, IL-4 and IL-13 in skin tissues and IgE and IL-4 in serum.

## 3. Discussion

Granulocytes such as mast cells and basophils play a critical role in IgE and IgE/antigen (Ag)-related allergic reactions [17]. RBL-2H3 cells are rat basophilic leukemia cells derived from mucosal mast cells and are commonly used for IgE-mediated degranulation studies [18]. The activation of granulocytes elicits degranulation, a significant step in allergic responses triggering the release of histamine or various cytokine cocktails, which are crucial in the progression and modulation of allergic inflammation [19,20]. Anaphylaxis, pruritus, and urticaria are induced in the early-phase reaction, and allergic march-related diseases such as AD, asthma, and allergic rhinitis appear in the late-phase reaction of allergic inflammation.

In this study, to examine the anti-allergic inflammatory potential of trifuhalol A, we first determined its effects on degranulation in anti-DNP IgE-treated RBL-2H3 cells. Trifuhalol A had concentration-dependent inhibitory effects on degranulation in RBL-2H3 cells. Since degranulation may be mediated by allergic or pseudoallergic routes, we evaluated the production of IgE and IL-33 in appropriate cell systems. The widely known IgE class switching condition is LPS and IL-4 stimulation in U266B1 cells. The application of trifuhalol A remarkably inhibited IgE production by the cells. IL-33 was induced by UVB and LL-37 treatment, an antimicrobial peptide derived from cathelicidin, in HaCaT cells, and the administration of trifuhalol A significantly ameliorated the production of IL-33. However, the effects of trifuhalol A on the expression of IL-4 were much lesser than its effects on degranulation. Trifuhalol A inhibited the production of IgE, which is responsible for the allergic route, and IL-33, which is involved in the pseudoallergic route. This means that trifuhalol A can alleviate degranulation through both routes.

IL-33 interacts co-operatively with IgE to promote the degranulation and induction of inflammatory cytokines in RBL-2H3 cells. The mechanisms involved in synergy between IL-33 and IgE in signaling can be explained at the level of TAK1 and MK2 [21,22]. TAK1 can be activated by an IL-33- and/or IgE-mediated response and stimulate IL-33, IL-6, and TNF-α production directly via the NF-κB pathway. Activated TAK1 can elicit the p38-MK2-PI3K/Akt pathway, and this axis also increases the production of related cytokines through NF-κB transactivation indirectly, and the degranulation of immune cells. Resveratrol inhibited the IL-33 and IgE-mediated activation of mast cells only through the MK2 pathway, whereas trifuhalol A simultaneously reduced it through both the TAK1 and MK2 pathways. The dysregulated activation of NF-κB contributes to the development of allergic inflammation-related diseases, such as asthma, allergic rhinitis, and dermatitis. Therefore, NF-κB is a key target for controlling allergic diseases, and for this, many attempts are being made to discover substances regulating the activation of NF-κB [23].

Anaphylaxis, part of the early-phase reactions, is the most severe form of hypersensitivity reaction, lasting minutes to hours after exposure. This response is triggered by the activation of basophils and mast cells, which provokes the release of mediators from secretory granules, such as histamine, carboxypeptidase A, tryptase, and proteoglycans [24]. The injection of Compound 48/80 stimulates mast cells to promote the secretion of histamine and can induce allergic reactions ranging from itching to anaphylaxis, depending on the dose [25,26,27]. The mortality rate was evaluated by inducing anaphylactic shock after injecting Compound 48/80 at a high concentration in a mouse model. Trifuhalol A reduced the mortality rate to 50% compared to the Compound 48/80-injected group. In addition, scratching behavior was measured in a mouse model in which itching was induced using a low concentration of Compound 48/80. Trifuhalol A decreased scratching. Thus, trifuhalol A can effectively improve death or itchiness caused by early-phase allergic reactions.

Atopic dermatitis (AD), one of the conditions related to the late-phase reaction, is a complex allergic inflammatory skin disease that negatively affects patients’ quality of life. AD is a common skin disease related to the Th2 response and the imbalance between Th1 and Th2 reactions [28]. The mediators involved in Th2 reactions include IL-4, IL-5, IL-13, IL-33, and IgE, in which IL-4 triggers dermatitis, similar to its effect in AD by inhibiting the production of IFN-gamma, thereby inducing an immune imbalance and increasing the synthesis of IgE [29,30].

It was previously reported that *Agarum cribrosum* had anti-inflammatory activity [10]. However, no systematic studies have been performed to evaluate its effects on allergic inflammation. Based on prescreening experiments, *Agarum cribrosum* and its compound trifuhalol A inhibited the IgE/DNP-BSA-driven release of β-hexosaminidase in RBL-2H3 cells. Hence, in this study, the therapeutic effect of trifuhalol A on allergic inflammation was evaluated.

Several allergens, such as ragweed and pollens of birch, mold, HDM, and grass, have been reported to exacerbate AD. HDM is often adopted to induce AD in a rodent model due to its high reproducibility and repeatability [31]. To induce lesions similar to AD, HDM prepared in the form of an ointment was administered to the back skin of NC/Nga mice. After inducing skin lesions with HDM in NC/Nga mice, the level of transepidermal water loss increased, and features such as water loss, erythema, and keratinization of the epidermis were observed. 

Exposure to allergy-causing substances triggers IgE production via IgE class switching in B cells and revitalizes mast cells; then, IgE and mast cells can trigger complex immune responses related to allergic symptoms [32]. Inflammation in AD is the process by which activated mast cells release inflammatory mediators [33]. Modulating the infiltration of mast cells from blood vessels into the dermal tissue or the degranulation of activated mast cells can reduce allergic inflammation in AD [34]. H&E and toluidine blue staining of the back skin of HDM-treated mice were conducted to examine epidermal and mast cell features. Based on the results of the skin tissue staining, the thickness of the epidermis and the number of infiltrating mast cells in the lesion area decreased in the trifuhalol A-treated group compared to the HDM group. 

One characteristic symptom of allergic inflammation is an increase in the plasma concentrations of IL-4 and IgE due to the activation of TH2 cells [35]. Pruritus is one of the most significant symptoms of AD induced by Th2 mediators, and it is an indispensable symptom of skin inflammation [36]. Hashimoto et al. reported a strong increase in scratching behaviors with intense dermatitis [37]. In this study, the HDM-treated NC/Nga mice exhibited strongly increased scratching behavior compared to the untreated group. However, this response was relieved by the administration of trifuhalol A. The elevated serum IL-4 and IgE levels in HDM-induced AD mice were markedly reduced by trifuhalol A treatment. As high IL-4 and IgE levels are hallmarks of several disorders associated with allergic disease, the serological and histological analysis markers improved by trifuhalol A intake provide a direct demonstration of the anti-allergic inflammatory effect of trifuhalol A in an animal model of AD.

TSLP has been linked to a variety of lesions, such as asthma, inflammatory arthritis, AD, and eczema [38]. A localized increase in TSLP in the skin is directly connected with Langerhans cell activation, leading to the differentiation of naive CD4 T cells into allergic CD4^+^ Th2 cells [39]. IL-33 enhances Th2-derived cytokine production from in vitro-polarized Th2 cells and inhibits Th1-derived cytokine production [40]. The administration of IL-33 elicits the expression of IL-4 and IL-13, recruits eosinophils (the main effector cells of allergic inflammation) to the lesion, and increases the levels of serum IgE in vivo [41]. This suggests that it may play a crucial role in the aggravation of allergic inflammation mediated by the activation of eosinophils [42]. Immunohistochemistry was conducted to confirm the protein expression level and location of TSLP and IL-33 in mouse skin. The expression levels of TSLP and IL-33 increased in the dorsal skin of the AD mouse model. However, the expression pattern of IL-33 and TSLP in the back skin of the mice treated with trifuhalol A was similar to that of the untreated group.

Type 2 immune responses are classically characterized by the differentiation of CD4^+^ T helper cells, the production of type 2 cytokines, IgE antibody class switching, and the recruitment of inflammatory effector cells [43,44]. Trifuhalol A is believed to inhibit differentiation into Th2 cells by reducing the expression of IL-4 through TSLP- or IL-33-mediated signaling cascades in vivo. Collectively, based on the above results, trifuhalol A is a potential therapeutic candidate to regulate allergic inflammation, such as asthma, allergic rhinitis, and atopic dermatitis.

## 4. Materials and Methods

### 4.1. Chemicals and Cells

Trifuhalol A that was isolated from *Agarum cribrosum* was donated by Professor Sang Moo Kim (Gangneung-Wonju National University, Gangneung, Korea). HDM (*Dermatophagoides farina*) extract ointment (Biostir AD) was obtained from Biostir Inc., (Osaka, Japan). ELISA kits for IL-4 and IgE were purchased from Invitrogen (Frederick, MD, USA). Power SYBR^®^ Green Master Mix was purchased from Applied Biosystems (Foster, CA, USA). RBL-2H3 rat basophilic leukemia cells were acquired from the Korean Cell Line Bank (KCLB No. 22256, Seoul, Korea) and cultured in DMEM supplemented with 10% fetal bovine serum (FBS) and antibiotics (100 U/mL of penicillin and 100 μg/mL of streptomycin; Invitrogen, CA, USA) at 37 °C in a humidified 5% CO_2_ atmosphere. The HaCaT immortalized human keratinocyte cell line (catalog no. T0020001; AddexBio, San Diego, CA, USA) was cultured in Dulbecco’s modified Eagle medium (DMEM; Gibco Lab., Grand Island, NY, USA) with 10% FBS and 1% (w/v) penicillin–streptomycin at 37 °C in a humidified CO_2_ incubator (95% air, 5% CO_2_). U266B1 human multiple myeloma cells (ATCC TIB196TM, Manassas, VA, USA) were cultured in RPMI 1640 medium supplemented with 10% FBS, 1 mM of sodium pyruvate, 2 mM of L-glutamine, 100 U of penicillin and 50 μg/mL of streptomycin at 37 °C in a 5% CO_2_ incubator.

### 4.2. Animals

Six-week-old female NC/Nga mice were procured from Japan SLC, Inc., (Shizoka, Japan). ICR mice (5 or 8 weeks old) were purchased from Orient Bio (Seongnam, Korea). All animals were housed in wire cages at 20–22 °C under a relative humidity of 50 ± 10% and 12 h illumination (06:00–18:00). All experiments were performed according to procedures approved by the Korea Institute of Science and Technology (KIST) Institutional Animal Care and Use Committee (IRB code No. 2020-001; KIST, Seoul, Korea).

### 4.3. β-Hexosaminidase Release from RBL-2H3 Cells

RBL-2H3 cells were seeded into 24-well plates, sensitized with anti-DNP-IgE (50 or 100 ng/mL) and /or IL-33 (70 pg/mL) for 4 h after seeding and incubated for another 4 h. After washing with Siraganian buffer (119 mM of NaCl, 5 mM of KCl, 1 mM of ${\mathrm{CaCl}}_{2}$, 0.4 mM of ${\mathrm{MgCl}}_{2}$, 25 mM of piperazine-N,N0-bis(2-ethanesulfonic acid) (PIPES), 5.6 mM of glucose, pH 7.2), the cells were incubated in the buffer for 10 min, treated with DMSO or trifuhalol A (10, 30, 100 μM) for 10 min, and treated then with DNP-BSA antigen (3 ng/mL or 10 μg/mL) for 20 min to stimulate degranulation. Supernatants were transferred to a 96-well plate and incubated with substrate (1 mM 4-nitrophenyl-N-acetyl-β-D-glucosaminide in 0.1 M of citrate buffer) at 37 °C for 3 h. The reaction was terminated by adding 200 µL of stop solution (0.1 M of ${\mathrm{Na}}_{2}{\mathrm{CO}}_{3}$/$N_{a}{\mathrm{HCO}}_{3}$, pH 10.0). The absorbance of each reaction was measured using an M1000 (TECAN, Salzburg, Austria) microplate reader at 405 nm.

### 4.4. Detection of IL-33 in UVB-Irradiated and LL-37-Treated HaCaT cells

The HaCaT cells were treated with 2 μM of LL-37 (Cayman Chemical, Ann Arbor, MI, USA) for 24 h prior to exposure to UVB (lamp Model G15T8E; 307 nm; Sankyo Denki, Japan) at an energy of 15 mJ/cm^2^, and trifuhalol A was administered (10, 30, and 10 μM). After 8 h, the cells were harvested to synthesize cDNA, and the level of IL-33 mRNA was measured using a quantitative real-time PCR (RT–qPCR).

### 4.5. Assay of IgE from U266B1 Cells

The U266B1 cells (1 × 10^6^ cells/well) were stimulated with IL-4 (5 ng/mL), LPS (10 μg/mL) and trifuhalol A (10, 30, 100 μM) for 72 h. The supernatants were harvested for IgE assays using the Human IgE Uncoated ELISA Kit (Invitrogen).

### 4.6. Luciferase Reporter Gene Assay of NF-κB in RBL-2H3 Cells

The RBL-2H3 cells were inoculated into a 100 mm^2^ culture dish and cultured for 20 h before transfection. The cells were co-transfected with NF-κB-Luc reporter plasmid and control plasmid pRL-SV-40 using Lipofectamine 2000 (Invitrogen) and cultured for 24 h. The transfected RBL-2H3 cells were seeded into 24-well plates and incubated for 20 h. Sensitization was performed using anti-DNP-IgE (50 ng/mL) for 4 h after seeding. After washing with Siraganian buffer (119 mM of NaCl, 5 mM of KCl, 1 mM of ${\mathrm{CaCl}}_{2}$, 0.4 mM of ${\mathrm{MgCl}}_{2}$, 25 mM of piperazine-N,N0-bis(2-ethanesulfonic acid) (PIPES), 5.6 mM of glucose, pH 7.2), the cells were incubated in the buffer for 10 min, treated with DMSO or trifuhalol A (100 μM) for 20 min, and then treated with DNP-BSA antigen (3 ng/mL) and IL-33 (70 pg/mL) for 30 min to stimulate degranulation. After 3 h, lysis was performed using passive lysis buffer (Promega, Madison, WI, USA). Luciferase assays were performed using the Dual-Luciferase^®^ Reporter Assay System (Promega), according to the manufacturer’s instructions. Relative luciferase activities were calculated with respect to Renilla luciferase activities.

### 4.7. Quantitative Real-Time PCR

Total RNA was isolated using an easy-BLUE^TM^ total RNA extraction Kit (iNTRON Biotechnology, Seoul, Korea). One milligram of the extracted RNA was reverse-transcribed using a RevertAid first strand cDNA synthesis kit (Thermo Fisher Scientific, Bremen, Germany), according to the manufacturer’s instructions. The following primers were selected: IL-4, forward 5′-CAG GTC AAC ACC ACG GAG AA-3′ and reverse 5′-ACA TCT CGG TGC ATG GAG TC-3′; IL-13, forward 5′-AGC TGA GCA ACA TCA CAC AAG AC-3′ and reverse 5′-GGC CAG GTC CAC ACT CCA T-3′; IL-33, forward 5′-AGC CTT GTG TTT CAA GCT GG -3′ and reverse 5′-ATG GAG CTC CAC AGA GTG TTC-3′; GAPDH, forward 5′-ACC ACA GTC CAT GCC ATC AC-3′ and reverse 5′-CCA CCA CCC TGT TGC TGT A-3′; IL-6, forward 5′-TAC CCC AAC TTC CAA TGC TC-3′ and reverse 5′-GGT TTG CCG AGT AGA CCT CA-3′; TNF-α, forward 5′-CAG CAG ATG GGC TGT ACC TT-3′ and reverse 5′-TGT GGG TGA GGA GCA CAT AG-3′. Data analyses were performed on 7500 System SDS software version 1.3.1 (Applied Biosystems, Walthan, MA, USA). The PCR settings were as follows: the initial denaturation for 15 s at 95 °C was followed by amplification for 40 cycles at 95 °C for 30 s and at 60 °C for 30 s, and a subsequent melting curve analysis was performed by increasing the temperature from 60 °C to 95 °C.

### 4.8. Compound 48/80-Induced Systemic Anaphylactic Reaction

Mice (8 weeks old, male) were administered an intraperitoneal injection of the mast cell degranulator Compound 48/80 (10 mg/kg). Trifuhalol A (200 mg/kg) dissolved in saline was orally administered 1 h before the Compound 48/80 injection. Mortality was monitored over 60 min after the induction of the anaphylactic reaction.

### 4.9. Compound 48/80-Induced Scratching Behavior Tests

Mice (5 weeks old, male) were subcutaneously administered Compound 48/80 (50 μg/kg) on the back of the neck. Trifuhalol A (20 mg/kg) in saline was administered orally 1 h before the injection of Compound 48/80. Scratching behavior was monitored for 1 h after the injection of Compound 48/80. One instance of scratching was recorded when the mouse scratched 3–4 times consecutively using a hind paw.

### 4.10. Induction of AD by HDM in NC/Nga Mice

Before applying HDM, the fur was shaved from behind the ears and off the back skin of the mice. In addition, a depilatory cream was used to completely remove the hair. One hundred milligrams of HDM was evenly applied to the back skin and both surfaces of the ears twice per week for 4 weeks. Trifuhalol A (20 mg/kg) was dissolved in saline and administered orally 5 times per week. At the end of the 4th week, the mice were sacrificed, and their dorsal skins and blood samples were collected for further investigation. The back skins were stored at −80 °C until RNA isolation and analysis and immediately fixed in 10% formalin for a histological analysis.

### 4.11. Measurement of Scratching Behavior Tests in NC/Nga Mice

Natural scratching by NC/Nga mice was measured for 1 h (01:00~02:00). All scratching behaviors were documented with a video recording camera (Mi Home Security Camera; Xiaomi, Shanghai, China). The counting and analysis were performed by an experimenter who was unaware of the treatment conditions.

### 4.12. Evaluation of the Skin Lesion (Dermatitis Score)

The severity of dermatitis induced by HDM treatment was evaluated immediately before sacrifice. The development of scarring/dryness, erythema/hemorrhage, edema, and excoriation/erosion was scored on a scale of 0 to 3 (0, none; 1, mild; 2, moderate; and 3, severe). The sum of the scores of each item was used as the dermatitis score.

### 4.13. Histological Examination

For the histological analysis, the dorsal skins of the NC/Nga mice were fixed in 10% neutral buffered formalin for 24 h and processed for paraffin wax embedding. The embedded tissues were sectioned at 2–3 mm. Hematoxylin and eosin (H&E) and toluidine blue staining were performed to measure changes in the thicknesses of the epidermis and the number of mast cells. Thestained tissues were analyzed using an optical microscope (Olympus CX31/BX51; Olympus Optical Co., Tokyo, Japan) and a fluorescence microscope (TE2000-U; Nikon Instruments Inc., Melville, NY, USA).

### 4.14. Measurement of Total Serum IL-4 and IgE Levels

Blood samples were collected from the abdominal aorta of the NC/Nga mice and then centrifuged at 8000 rpm for 15 min at 4 ℃ to obtain the serum. Total serum IL-4 (Mouse IL-4 ELISA Ready-SET-Go!, eBioscience, San Diego, CA, USA) and IgE (Mouse IgE ELISA Ready-SET-Go!, eBioscience) levels were measured using enzyme-linked immunosorbent assay (ELISA) kits.

### 4.15. Measurement of Cytokine Levels

Supernatants from the RBL-2H3 cells were obtained and subjected to an ELISA to determine the levels of IL-6 (Rat IL-6 Ezway Cytokine ELISA Kit, LABISKOMA, Seoul, Korea) and TNF-α (Rat TNF-α Ezway Cytokine ELISA Kit, LABISKOMA).

### 4.16. Immunohistochemistry

Mouse skin samples were embedded in paraffin and fixed in 10% formalin. For antigen retrieval, the paraffin-embedded sections were boiled in 1 mM of citrate buffer, pH 6, for 20 min. The mouse skin tissues were stained for TSLP (1:200 Abcam, Cambridge, UK) and IL-33 (1:200 Invitrogen, Waltham, CA, USA). The tissue sections were then incubated for 30 min with a biotinylated goat anti-rabbit secondary antibody (Vector Labs, Burlingame, CA, USA). Antibody binding was visualized using a DAB Peroxidase Substrate Kit (Vector Labs). After the substrate reaction, the tissue sections were washed with water and counterstained with hematoxylin. The tissue sections were photographed using the Nikon Eclipse E200 System (Nikon, Tokyo, Japan).

### 4.17. Statistical Analysis

All the results are expressed as the means ± SD. The significance of differences between the mean values in the two groups was analyzed using a one-way analysis of variance (ANOVA). A value of *p* < 0.05 was considered statistically significant.

## 5. Conclusions

In conclusion, trifuhalol A can ameliorate the symptoms of allergic inflammation through the dual inhibition of TAK1 and MK2 mediated by IL-33 and IgE. In addition, the inhibition of degranulation by trifuhalol A can improve the anaphylactic shock and itching that appear as an early-phase reaction and allergic diseases such as atopic dermatitis as a late-phase reaction. Therefore, trifuhalol A is a new drug candidate for the regulation of allergic inflammation.

## Figures and Tables

**Figure 1 ijms-23-10163-f001:**
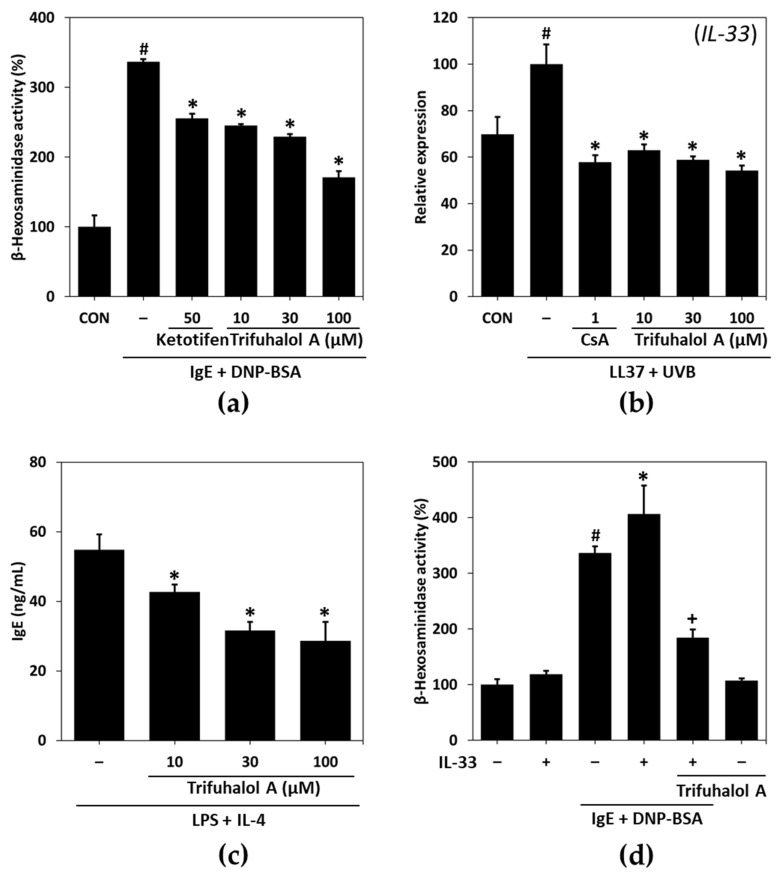
Effects of trifuhalol A on allergic inflammation-related mediators. (**a**) RBL-2H3 cells were sensitized with anti-DNP IgE (100 ng/mL) and treated with the indicated concentrations of trifuhalol A and DNP-BSA (1 μg/mL). The β-hexosaminidase release was calculated relative to the control (CON). Each bar represents the mean ± S.D. of the duplicates. ^#^
*p* < 0.05 vs. the control group; * *p* < 0.05 vs. the anti-DNP IgE + DNP-BSA-treated group. (**b**) Representative data of qPCR analysis of IL-33 mRNA normalized to β-actin in LL37 (2 μM) + UVB (15 mJ/cm^2^)-treated HaCaT cells. HaCaT cells were treated with trifuhalol A in combination with LL37/UVB for 24 h to measure IL-33 expression. Each bar represents the mean ± SD of two experiments. ^#^
*p* < 0.05 vs. control; * *p* < 0.05 vs. LL37/UVB-treated group. CsA; 1 μM cyclosporine A. (**c**) U266B1 cells were stimulated with LPS (4 μg/mL) + IL-4 (5 ng/mL) in the presence or absence of trifuhalol A for 72 h, and IgE levels in the culture supernatants were measured using ELISA. Each bar represents the mean ± S.D. of two experiments. * *p* < 0.05 vs. the LPS + IL-4-treated group. (**d**) RBL-2H3 cells were sensitized with anti-DNP IgE (50 ng/mL) and IL-33 (70 pg/mL) and treated with trifuhalol A (100 uM) and DNP-BSA (3 ng/mL). The β-hexosaminidase release was calculated relative to the control (CON). Each bar represents the mean ± S.D. of the duplicates. ^#^
*p* < 0.05 vs. the control group; * *p* < 0.05 vs. the anti-DNP-BSA-treated group; ^+^
*p* < 0.05 vs. the anti-DNP-BSA-IL-33-treated group.

**Figure 2 ijms-23-10163-f002:**
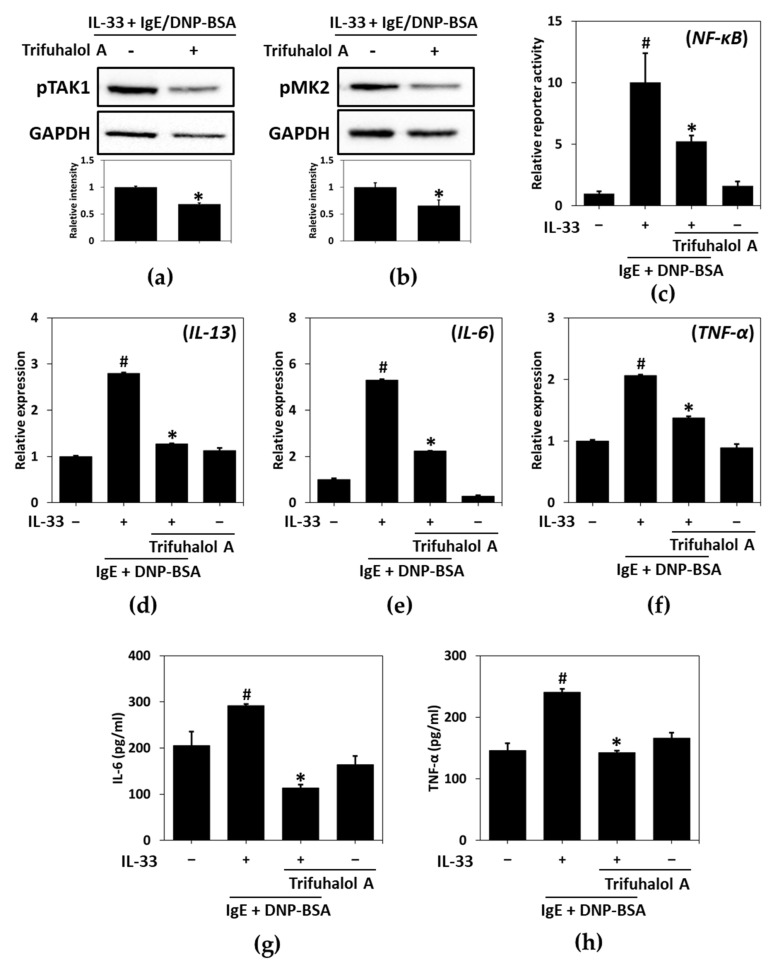
Effects of trifuhalol A on IL-33- and IgE-induced immune activation. Phosphorylation patterns of TAK1 (**a**) and MK2 (**b**) in RBL-2H3 cells in response to IL-33 (70 pg/mL) and IgE (50 ng/mL) in the presence or absence of trifuhalol A (100 μM). * *p* < 0.05 vs. IL-33 + IgE + DNP-BSA treated group. Each bar represents the mean ± S.D. * *p* < 0.05 vs. IL-33 + IgE/DNP-BSA-treated group. (**c**) Luciferase assay of NF-κB transactivation in RBL-2H3 cells treated with IL-33 and IgE in the presence or absence of trifuhalol A (100 μM). Transcriptional levels of IL-13 (**d**), IL-6 (**e**) and TNF-α (**f**) stimulated by IL-33 and IgE for 4 h in the presence or absence of 100 μM of trifuhalol A. Translational levels of IL-6 (**g**) and TNF-α (**h**) stimulated by IL-33 and IgE for 4 h in the presence or absence of 100 μM of trifuhalol A. ^#^
*p* < 0.05 vs. control group; * *p* < 0.05 vs. IL-33 + IgE + DNP-BSA treated group.

**Figure 3 ijms-23-10163-f003:**
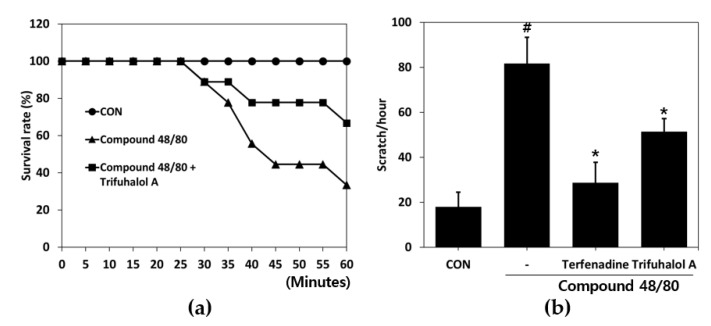
Effects of trifuhalol A on Compound 48/80-induced anaphylactic shock and scratching behavior in ICR mice. (**a**) Mice were administered trifuhalol A (200 mg/kg) for 1 h before Compound 48/80 injection (10 mg/kg i.p.) (n  =  9 per group). The survival rates of these mice were monitored for 1 h. (**b**) Mice were treated with or without trifuhalol A (20 mg/kg) or terfenadine (4 mg/kg) for 1 h before Compound 48/80 injection (50 μg/kg s.c.) (n = 3 per group). The scratching behavior was counted for 1 h after Compound 48/80 injection. Each bar represents the mean ± S.D. ^#^
*p* < 0.05 vs. control group; * *p* < 0.05 vs. Compound 48/80-treated group.

**Figure 4 ijms-23-10163-f004:**
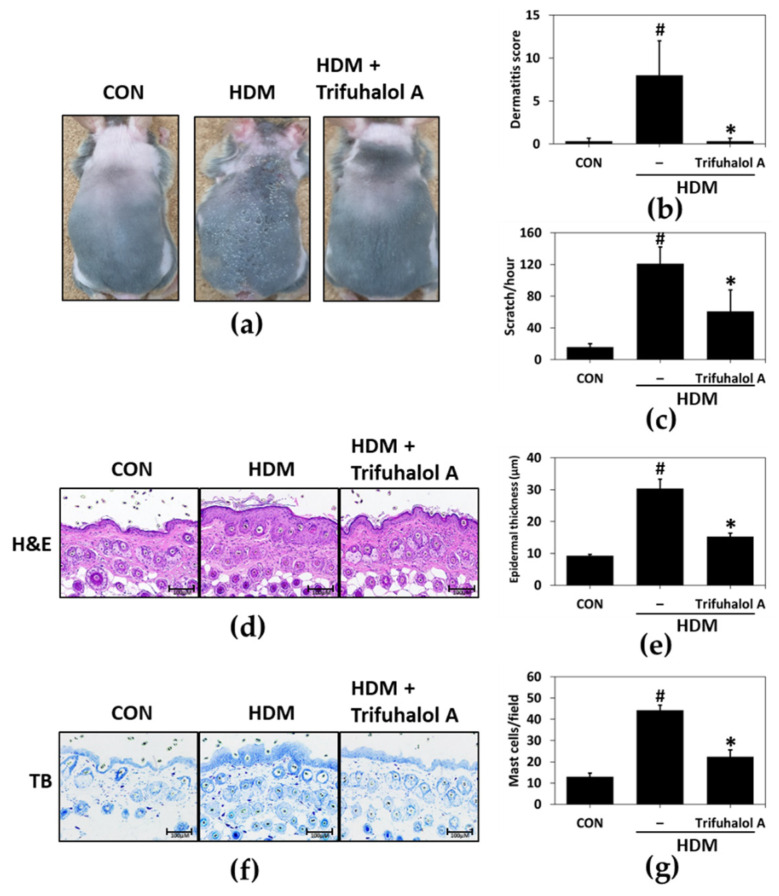
Effects of trifuhalol A on HDM-induced AD-like symptoms in the dorsal skin of NC/Nga mice. (**a**) Clinical features of trifuhalol A treatment. (n = 5 in each group). (**b**) The dermatitis score was evaluated by scoring the skin lesions according to symptoms such as erythema/hemorrhage, scarring, edema, and excoriation/erosion. Each bar represents the mean ± S.D. (n = 5 in each group). ^#^
*p* < 0.05 vs. control group; * *p* < 0.05 vs. HDM group. (**c**) Scratching behavior was observed for 1 h at the end of the experiment. Each bar represents the mean ± S.D. (n = 5 in each group). ^#^
*p* < 0.05 vs. control group; * *p* < 0.05 vs. HDM group. (**d**) Histopathological features induced by hematoxylin and eosin staining of dorsal skin lesions (magnification ×100) and (**e**) measured epidermal thicknesses. Each bar represents the mean ± S.D (n = 5 in each group). ^#^
*p* < 0.05 vs. control group; * *p* < 0.05 vs. HDM group. (**f**) Infiltrated mast cells were shown through toluidine blue staining of dorsal skin lesions (magnification ×100), and (**g**) the number of mast cells per dermis. Each bar represents the mean ± S.D (n = 5 in each group). ^#^
*p* < 0.05 vs. control group; * *p* < 0.05 vs. HDM group.

**Figure 5 ijms-23-10163-f005:**
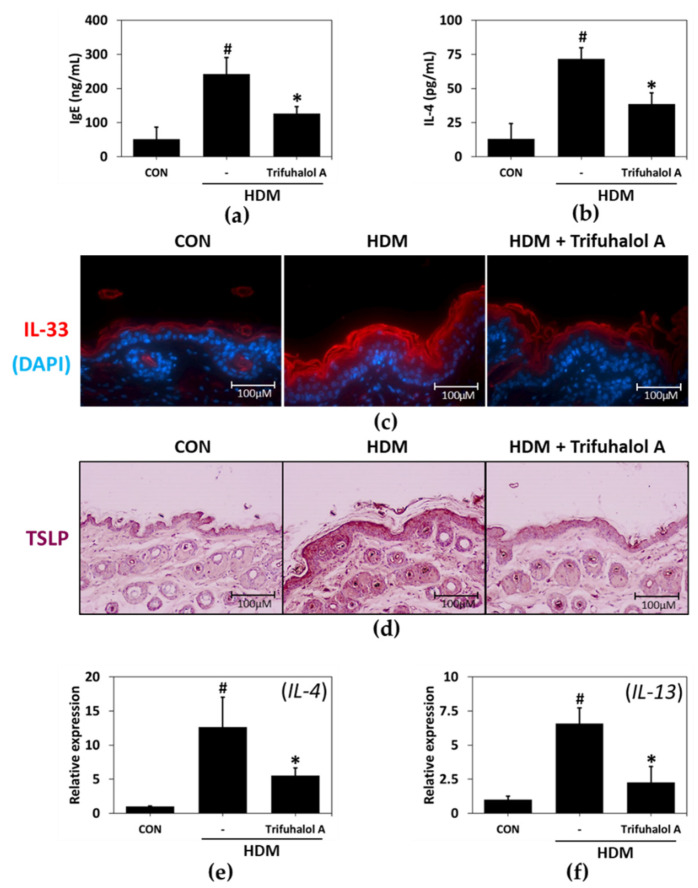
Effects of trifuhalol A on the serum and tissue factors related to allergic inflammation in HDM-induced AD-like NC/Nga mice. (**a**) Serum IgE and (**b**) IL-4 levels were measured using ELISA (n = 5 in each group). Immunohistochemistry was used to identify the expression level and sites of (**c**) IL-33 and (**d**) TSLP proteins in the skin (n = 5 in each group) (magnification ×100). Representative data of qPCR analysis of (**e**) IL-4 and (**f**) IL-13 mRNA normalized to GAPDH. Each bar represents the mean ± SE (n = 5 in each group). ^#^
*p* < 0.05 vs. control group; * *p* < 0.05 vs. HDM group.

## Data Availability

Not applicable.

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
