# Peer review of "Trifuhalol A Suppresses Allergic Inflammation through Dual Inhibition of TAK1 and MK2 Mediated by IgE and IL-33"

_ijms, 2022, doi:10.3390/ijms231710163_

Round 1

Reviewer 1 Report (Previous Reviewer 1)

The authors have addressed some comments, but still the following should be addressed:

1- The novelty of this study should be highlighted since there are many similar studies.

2- In the in vitro studies, the authors used 3 concentrations (10, 30 and 100 uM) of trifuhalol. What was the basis of selecting these specific concentrations? Does trifuhalol has any effect at doses lower than 10 uM?

3- In section 4.8, the authors administered Trifuhalol A (200 mg/kg); however, in experiments 4.9. and 4.10, 20 mg/kg was administered. This should be explained.

4- Regarding the doses of Trifuhalol A, in particular the 200 mg/kg dose, Why only one dose was applied? and is there any dose-dependent effect? What about lower and higher doses?!

5- The authors employed PCR for the assessment of cytokines; however, protein levels determined through ELISA or other blotting techniques are important.

6- The role of NF-kB should be more emphasized.

7- Is there any connection of the released cytokines with increased ROS levels?

Author Response

Response to Reviewer 1’s Comments

Dear Reviewer,

Thank you for your valuable comments.

We answered point-by point the reviewers’ specific comments and all changes are marked by highlighted in red color in the revised manuscripts.

Reviewer’s comments)

The authors have addressed some comments, but still the following should be addressed:

1- The novelty of this study should be highlighted since there are many similar studies.

Answer) In this study, we identified that trifuhalol A inhibits the production of IL-33 and IgE and reduces degranulation and activation by them. So, we next study the effects of trifuhalol A on the NF-κB and MAPK-activated protein kinase 2 (MK2) in RBL-2H3 cells. IL-33 and IgE augmented the phosphorylation of TGFβ-activated kinase 1 (TAK1) and MK2 leading to activation of NF-κB pathway to produce several cytokines such as IL-13, IL-6 and TNF-α. Trifuhalol A reduced IL-33- and IgE-induced phosphorylation of TAK1 and MK2 in RBL-2H3 cells (Figure 2a and b). In addition, NF-κB regulated cytokines associated with TAK1 and MK2, such as IL-13, IL-6 and TNF-α, also increased by IL-33 and IgE co-treatment and the up-regulated transcriptional level of them was declined by trifuhalol A. These results suggest that suppression of IL-33- and IgE-mediated activation of RBL-2H3 cells by trifuhalolA can be attributed to the inhibition of the TAK1 and MK2 pathway. Resveratrol is known to inhibit IL-33 and IgE-mediated activation of mast cells only through the MK2 pathway (Ref 22). So, our study is considered to have novelty.

2- In the in vitro studies, the authors used 3 concentrations (10, 30 and 100 uM) of trifuhalol. What was the basis of selecting these specific concentrations? Does trifuhalol has any effect at doses lower than 10 uM?

Answer) In our lab., the standard for judging that natural compounds are effective is 10 uM. Since weak efficacy was shown at this concentration, the efficacy was observed at a higher concentration to check the dose dependency. It is predicted to show a weak effect at doses lower than 10 uM.

3- In section 4.8, the authors administered Trifuhalol A (200 mg/kg); however, in experiments 4.9. and 4.10, 20 mg/kg was administered. This should be explained.

Answer) In our lab., the concentration showing efficacy in vivo is predicted to be 100 ~ 1,000 times higher than the concentration showing efficacy in vitro. Since the molecular weight of Trifuhalol A is 390.3, it was predicted that it would show efficacy at the concentrations between 10 (3.903 mg/ml) to 100 mM (39.03 mg/ml), which is 100 ~ 1,000-fold concentration of 100 uM. So, in this study, efficacy tests were performed at the concentrations of 20 mg/Kg corresponding to 500 times the effective dose in vitro in experiments 4.9 and 4.10. Fortunately, effects against itchiness and atopic dermatitis were well observed at the concentration used. Compound 48/80 was used to induce pruritus (4.9) or anaphylaxis (4.8). Compound 48/80 was injected to mice at a concentration of 50 ug/kg to induce itch and 10 mg/kg to induce anaphylaxis. In general, a high concentration of the test sample is used to measure the efficacy against anaphylaxis, and our compound also showed efficacy at high dose (200 mg/kg).

4- Regarding the doses of Trifuhalol A, in particular the 200 mg/kg dose, Why only one dose was applied? and is there any dose-dependent effect? What about lower and higher doses?!

Answer) Initially, efficacy tests for anaphylaxis were performed at 20 mg/kg as previous trials, but all animals used in the test died. So, the next tests were performed on two animals each at the concentrations of 50, 100, 200, 400, and 500 mg/Kg. At 50 and 100 mg/kg, all died, at 400 and 500 mg/kg, both survived, and at 200 mg/kg, one mouse survived. So, the test was carried out at 200 mg/Kg.

5- The authors employed PCR for the assessment of cytokines; however, protein levels determined through ELISA or other blotting techniques are important.

Answer) As your valuable comment, we measured the protein levels of cytokines (IL-6 and TNF-α) by ELISA and added them to the results section. (p 4,l147-148)

We tried to purchase ELISA kits for IL-13, IL-6 and TNF-α and obtained kits for rat IL-6, TNF-α and mouse IL-13. Protein levels for IL-6 and TNF-α are well accordance with qPCR data. The ELISA kit for rat IL-13 was out of stock in Korea, it takes several months to arrive and there are no users around, so the results could not be contained in this revision. But it is predicted to agree well with the data shown by qPCR. Because we tried to measure the protein level of IL-13 with a mouse IL-13 Ezway Cytokine ELISA Kit known to have cross-reactivity with rats and the concentration was very low but the pattern showed similar to the qPCR data. The data measured for IL-13 is attached below.

6- The role of NF-kB should be more emphasized.

Answer) As your valuable comment, we added a little more emphasis on the role of NF-kB to the discussion section. (Ref 23, p10, l296-299)

7- Is there any connection of the released cytokines with increased ROS levels?

Answer) There are many reports that ROS increases the release of pro-inflammatory cytokines. But, it seems to be very little connection between the released cytokines increased by IL-33 and IgE co-treatment and increased ROS levels, in this study. (Ref 21 and 22)

Reviewer 2 Report (Previous Reviewer 2)

Comments to the Author

In the study, the authors found that Trifuhalol A inhibits allergic inflammation via the suppression of immune cell degranulation mediated by IgE and IL-33 could serve as therapeutic target for treatment of allergic inflammation. IgE and IL-33 is a well known biomarkers and future therapeutic targets for allergic and immunological disorders. The study is well-established projects and the results are less interesting. However, several major rejection issues should be addressed more clearly.

1. Does the IgE and IL-33 is ranked as the top gene in the unregulated genes in the treatment Trifuhalol A sensitivity immune cell degranulation by sequencing analysis?

2. The authors should performed the different experiments through down-regulation or over-expression of IgE and IL-33 to verify the hypotheses of Trifuhalol A treatment sensitivity. The title is not accurate and should be modified.

3. There are a lot of English grammar errors in the manuscript.

Author Response

Dear Reviewer,

Thank you for your valuable comments.

We answered point-by point the reviewers’ specific comments and all changes are marked by highlighted in red color in the revised manuscripts.

Reviewer’s comments)

In the study, the authors found that Trifuhalol A inhibits allergic inflammation via the suppression of immune cell degranulation mediated by IgE and IL-33 could serve as therapeutic target for treatment of allergic inflammation. IgE and IL-33 is a well known biomarkers and future therapeutic targets for allergic and immunological disorders. The study is well-established projects and the results are less interesting. However, several major rejection issues should be addressed more clearly.

  1. Does the IgE and IL-33 is ranked as the top gene in the unregulated genes in the treatment Trifuhalol A sensitivity immune cell degranulation by sequencing analysis?

Answer) In several previous publishes, IgE and IL-33 were reported as the first molecules involved in the degranulation and activation of immune cells. In this manuscript, since trifuhalol A inhibits the class-switch of IgE and the biosynthesis of IL-33, IgE and IL-33 seem to be the first signals that trifuhalol A acts among a series of events occurring in the degranulation and activation of granulocytes. (Ref 21 and 22)

  1. The authors should performed the different experiments through down-regulation or over-expression of IgE and IL-33 to verify the hypotheses of Trifuhalol A treatment sensitivity. The title is not accurate and should be modified.

Answer) As your valuable comment, we have changed the title of this manuscript from “Trifuhalol A inhibits allergic inflammation via the suppression of immune cell degranulation mediated by IgE and IL-33” to “Trifuhalol A suppresses allergic inflammation through dual inhibition of TAK1 and MK2 mediated by IgE and IL-33”.

  1. There are a lot of English grammar errors in the manuscript.

Answer) Before submitting our manuscript, English sentences were corrected by a qualified native English speaking editor at American Journal Experts (Verification code 53EC-2294-4FAE-5749-78FC).

Round 2

Reviewer 1 Report (Previous Reviewer 1)

The authors have addressed most of my comments and explained the non-addressed comments. I hope they consider the raised concerns in their future work.

Author Response

I appreciate your valuable comments and thank you again . English sentences and grammars were corrected by a qualified native English speaking editor at MDPI (English-49813).

Reviewer 2 Report (Previous Reviewer 2)

The authors have addressed all my concerns and I thank them for that. I have no further request and I think this manuscript is now acceptable for publication. I just suggest a careful review of the text for correcting some typos.

Author Response

I appreciate your valuable comments. English sentences and grammars were corrected by a qualified native English speaking editor at MDPI (English-49813).

This manuscript is a resubmission of an earlier submission. The following is a list of the peer review reports and author responses from that submission.

Round 1

Reviewer 1 Report

The authors investigated the inhibitory effect of the phlorotannin trifuhalol on degranulation of immune cells and the release of IgE and IL-33 from keratinocytes and B cells, and employed Compound 48/80-induced systemic anaphylaxis and house dust mite (HDM)-induced atopic dermatitis (AD) mouse models.

Information on the isolation and characterization of trifuhalol should be added to the manuscript (even if it has been donated by a collaborator).

The authors employed PCT for the assessment of cytokines; however, protein levels determined through ELISA or other blotting techniques are important.

In the in vitro studies, the authors used 3 concentrations (10, 30 and 100 uM) of trifuhalol. What was the basis of selecting these specific concentrations? Does trifuhalol has any effect at doses lower than 10 uM?

More or less similar, 200 mg/kg dose was used in the in vivo studies. Why and is there any dose-dependent effect? What about lower and higher doses?!

Histology and IHC pictures should be enlarged to reflect the details.

What is the mechanism(s) underlying the effect of trifuhalol on the cytokines? Is there sny involvement of NF-kB or other transcription factors?

Is there any connection of the released cytokines with increased ROS levels?

Although substantial work has been done in this manuscript, the mechanistic explanation is still lacking.

Reviewer 2 Report

This original article study is largely confirmatory of a previously published study by J Food Biochem. 2021 Apr;45(4):e13659.; J Photochem Photobiol B. 2016 Sep;162:100-105.; May 2019Natural Product Communications 14(5):1934578X1984979; PLoS One. 2012;7(2):e31145. doi: 10.1371/journal.pone.0031145.; Mar Drugs. 2018 Aug 2;16(8):267.  Preclinical studies underscore clearly the understanding of the mechanisms by which orally administered Phlorotannins for suppress chemical mediator release and cyclooxygenase-2 signaling pathway and well established the anti-Inflammatory activity of the Phlorotannin Trifuhalol A on allergic inflammation functions and therefore lacks significant novelty of this original study.